# Analysis of the Changes in Diversity of Culturable Bacteria in Different Niches of Mulberry Fields and Assessment of Their Plant Growth-Promoting Potential

**DOI:** 10.3390/microorganisms13051012

**Published:** 2025-04-28

**Authors:** Weifu Liu, Ting Yuan, Mengya Wang, Jiping Liu

**Affiliations:** Guangdong Provincial Key Lab of Agro-Animal Genomics and Molecular Breeding, College of Animal Science, South China Agricultural University, Regional Sericulture Training Center for Asia-Pacific, Guangzhou 510642, China; liuweifu2025@163.com (W.L.); yuanting361@163.com (T.Y.); wangmengya0910@163.com (M.W.)

**Keywords:** mulberry, rhizosphere soil bacteria, endophytic bacteria, intestinal bacteria, biological control, sericulture

## Abstract

Microorganisms play a crucial role in agricultural systems. The use of plant growth-promoting bacteria (PGPB) to enhance agricultural production in a sustainable and environmentally friendly manner has been widely recognized as a key technology for the future. In this study, we analyzed the diversity changes of bacteria in different ecological niches of mulberry fields based on culture-dependent methods, and we further evaluated their antibacterial and plant growth-promoting (PGP) activities. A total of 346 cultivable bacteria belonging to 30 genera were isolated from mulberry rhizosphere soil, mulberry plants and silkworm intestines, among which the dominant genera were *Bacillus*, *Pseudomonas*, and *Enterobacter*. The bacterial communities in the mulberry rhizosphere soil were more diverse than those in the mulberry endophytes and in the silkworm intestines. The antibacterial test showed that 30 bacteria exhibited antibacterial activity against the plant pathogen *Ralstonia solanacearum*. PGP trait assays indicated that 58 bacteria were capable of nitrogen fixation, phosphate solubilization, potassium release and siderophore production simultaneously. The screened functional strains promoted the growth of mulberry saplings. The results of this study highlight new findings on the application of silkworm intestinal bacteria in PGPB.

## 1. Introduction

Green development has emerged as a pivotal global concern. The green development of agriculture is now seen as an essential component of the agricultural modernization. The application of fertilizers and bactericides is a commonly employed practice for achieving high crop yields and reducing the incidence of diseases. The long-term and excessive use of chemical fertilizers and chemical bactericides can lead to soil and water pollution, disrupt the microbial community structure, lead to a decline in soil fertility and ecosystem productivity, increase crop susceptibility to diseases, and impede the green development of agriculture [1]. The use of plant growth-promoting bacteria (PGPB) has seen an increase in its application in agricultural research. The utilization of PGPB to enhance agricultural production in a sustainable and environmentally friendly manner has been widely regarded as a key technology for the future [2]. PGPB can transform inaccessible nutrients into available forms through biological nitrogen fixation, phosphate solubilization, potassium release, and siderophore production, thereby facilitating plant nutrient absorption and promoting plant growth [3,4,5]. Besides nutrient acquisition, PGPB can suppress pathogens harmful to plant growth and trigger defense responses in plants [6,7,8]. Moreover, PGPB are capable of biosynthesizing auxins, like indole-3-acetic acid (IAA), which stimulate plant growth [4,9,10]. PGPB have the potential to be developed into microbial fertilizers and biological control agents (BCAs), replacing chemical fertilizers and chemical bactericides to establish environmentally friendly and sustainable agricultural systems [11,12,13,14].

Mulberry, an agriculturally and economically significant crop, is extensively cultivated in subtropical and temperate regions [15]. Mulberry leaves are extensively used as a feed resource for the domestic silkworms [16]. Mulberry is now used as a supplement in animal feed [17]. Additionally, mulberry is widely used in medicine, food, and healthcare [18,19,20]. Soil compaction is a significant challenge faced by modern agriculture, and mulberry fields are no exception. The insufficient application of organic fertilizers and the long-term overuse of chemical fertilizers have further exacerbated the degradation of the soil in mulberry fields, impeding their sustainable development. As a crucial component of a mulberry field, PGPB play a vital role in facilitating nutrient circulation. In recent years, several studies have revealed that PGPB in mulberry fields can antagonize mulberry pathogens, thereby exerting a preventive and controlling effect on mulberry diseases [21,22,23]. Moreover, these bacteria possess certain characteristics that promote plant growth. It is anticipated that the use of PGPB can help reverse soil deterioration in mulberry fields, enabling them to achieve sustainable development.

PGPB have a high potential to be used as microbial fertilizers and BCAs for application in mulberry fields. Understanding the bacterial community structure and composition diversity within mulberry fields is beneficial for the development and management of mulberry field microbial fertilizers and BCAs [24]. The composition of endophytic bacteria differs among various mulberry varieties [25]. Seasonal changes can also influence the composition of mulberry endophytic bacteria [26]. In addition, mulberry disease can also lead to changes in the diversity of bacteria within mulberry [27]. Mulberry rhizosphere soil bacteria, mulberry endophytic bacteria, and silkworm intestinal bacteria are the three important bacterial communities of mulberry fields. There is a certain connection and a dynamic flow relationship among these three types of bacteria. The mulberry rhizosphere soil bacteria can be transferred from the mulberry rhizosphere soil to the mulberry through colonization, and they then exist as mulberry endophytic bacteria. When mulberry leaves are used to feed silkworms, the mulberry endophytic bacteria can be transferred from the mulberry leaves to the silkworm intestines, and they then become the silkworm intestinal bacteria. When silkworm dung is returned to the field, the silkworm intestinal bacteria are transferred from the silkworm intestines to the soil, and they then become the mulberry rhizosphere soil bacteria. It is a very interesting cycle, but we do not know much about it. Investigating the connections between mulberry rhizosphere soil bacteria, mulberry endophytic bacteria, and silkworm intestinal bacteria, revealing the ecological interaction mechanism of microorganisms in the mulberry field system, and identifying and utilizing the core microbial community are expected to promote the sustainable development of the sericulture industry. However, the structural diversity and dynamic changes among these three bacterial groups have rarely been reported.

Research on mulberry growth-promoting bacteria and biological control bacteria has predominantly focused on mulberry endophytic bacteria [21,23]. Comparatively, investigation on mulberry rhizosphere soil bacteria and silkworm intestinal bacteria as mulberry growth-promoting and biological control agents is scarce. Mulberry soil-borne diseases, like mulberry bacterial wilt, significantly endanger the health of mulberry plants [27,28]. Consequently, screening mulberry rhizosphere soil bacteria for the prevention and control of mulberry diseases may hold greater significance. Additionally, since PGPB, such as plant growth-promoting rhizobacteria (PGPR), primarily function in the plant rhizosphere soil, it may be more rational to screen for mulberry growth-promoting bacteria among the mulberry rhizosphere soil bacteria. Silkworm intestinal bacteria originate from silkworm intestines, and those in healthy silkworm intestines can be regarded as safe for silkworms [29]. For the safety of silkworms, screening mulberry growth-promoting bacteria and biological control bacteria from silkworm intestinal bacteria is also a feasible approach. However, few silkworm intestinal bacteria have been utilized to promote mulberry growth and control mulberry diseases. The objectives of this study were (1) to isolate cultivable bacteria from the mulberry rhizosphere soil, mulberry plants, and silkworm intestines, and then analyze the diversity of these bacteria communities; (2) to analyze and screen mulberry growth-promoting bacteria through antibacterial test, nitrogen fixation, phosphate solubilization, potassium release and siderophore production; and (3) to evaluate the effect of bacteria with potential applications as microbial fertilizers and microbial control agents on the development and application in promoting mulberry growth. Ultimately, the aim of this research was to establish a strain reservoir of mulberry field bacteria and further obtain strain resources with potential applications in mulberry microbial fertilizers and microbial control agents.

## 2. Materials and Methods

### 2.1. Collection of the Samples

In this study, healthy mulberry rhizosphere soil, stems and leaves were collected from six mulberry fields (Appendix A) in Guangdong Province. Subsequently, the samples were transported back to the laboratory and temporarily stored in a 4 °C refrigerator. Silkworms were reared under standard rearing conditions until they reached the fifth instar. Then, healthy silkworms from eight different varieties (Qiu Hua × Ping 30, Liang Guang No. 29327532, Furong, Xianghui, 932 × Furong, 7532 × Xianghui) were selected. After fasting these silkworms for 24 h, they were used for the experiment.

### 2.2. Isolation of Mulberry Field Bacteria

For the mulberry rhizosphere soil samples, 10 g of the soil was weighed and transferred into a triangular flask containing glass beads and 100 mL of sterile water. Subsequently, the flask was placed in a shaker operating at 180 r/min and 28 °C for 30 min to prepare the mother bacterial suspension (base solution) of mulberry rhizosphere soil bacteria.

For the mulberry root, stem and leaf samples, the surface sediments on the mulberry roots, stems, and leaves were first washed with clean water. Then, the samples were cut into appropriate sizes and soaked in a 75% ethanol/water (*v*/*v*) solution for disinfection. After disinfection, the samples were rinsed three times with sterile water to remove the residual ethanol on their surfaces. Then, 100 μL of the last-rinsed water was added to the center of Lysogeny Broth (LB) agar plates (Guangdong Huankai Microbial Technology Co., Ltd., Guangzhou, China) and Nutrient Agar (NA) plates (Guangdong Huankai Microbial Technology Co., Ltd., Guangzhou, China) and evenly spread on the plate surfaces using a sterile glass spreading rod. The plates were sealed and incubated in a 28 °C constant-temperature incubator. The absence of colony growth on the plates indicated the effectiveness of the disinfection. The sterilized roots, stems, and leaves were crushed and placed into a triangular flask containing sterile water. The flask was then placed in a shaker operating at 180 r/min and 28 °C for 30 min to prepare the mother bacterial suspension (base solution) of mulberry endophytic bacteria.

For the silkworm samples, healthy silkworms were fasted for 24 h. Their surfaces were then rinsed three times with sterile water and then soaked in a 75% ethanol/water (*v*/*v*) solution for disinfection. After disinfection, the silkworms were rinsed three more times with sterile water to remove the residual ethanol on their surfaces. Under sterile conditions, the silkworm intestines were exposed, cut off, and placed into a triangular flask containing sterile water. The flask was then placed in a shaker operating at 180 r/min and 28 °C for 30 min to prepare the mother bacterial suspension (base solution) of silkworm intestinal bacteria.

All the bacterial suspensions mentioned above were serially diluted to concentrations of 10^−1^, 10^−2^, 10^−3^, 10^−4^, 10^−5^, 10^−6^, 10^−7^, and 10^−8^. A volume of 100 μL from each dilution was added to the center of the NA plates and LB agar plates. The inoculum was then spread evenly on the surfaces of the plates using a sterile glass spreading rod. The plates were sealed and incubated in a 28 °C constant-temperature incubator. Once colonies had grown on the plates, the plates with 30–300 colonies were selected. Single colonies of different morphologies were picked and streaked onto new NA plates and LB agar plates. Each colony was purified through 7 generations, and all the purified strains were stored in 30% glycerol at −80  ℃.

### 2.3. Identification of the Bacteria

Identification of the bacteria was based on the analysis of the 16S rRNA gene sequencing using the universal primers 27F/1492R [30]. The total DNA of each strain was extracted using the Ezup Column Bacteria Genomic DNA Purification Kit (Sangon Biotechnology Co., Ltd., Shanghai, China), following the manufacturer’s instructions precisely. The 16S rRNA gene of DNA from all the isolated and purified strains was amplified by PCR using the 25 μL reaction system, performed as follows: one cycle of 95 °C for 5 min; followed by 33 cycles of 94 °C for 1 min, 56 °C for 1 min, and 72 °C for 2 min; and a final extension at 72 °C for 10 min. The PCR amplification products were detected by 1.2% agarose gel electrophoresis and the qualified PCR amplification products were sequenced by the Sanger method at Sangon Biotechnology Co., Ltd., Shanghai, China. The generated sequences were aligned using BioEdit software version 7.0 and then subjected to analysis by the Basic Local Alignment Search Tool (BLAST+ 2.16.0) search program of the NCBI database (National Center for Biotechnology Information, https://blast.ncbi.nlm.nih.gov/Blast.cgi (accessed on 26 February 2024) to determine the sequence homology with closely related organisms [31]. All the bacterial isolates were classified to the genus level based on the closest bacterial information.

### 2.4. Determination of the Antimicrobial Activity of the Bacteria

The antimicrobial activity of the mulberry rhizosphere soil bacteria, mulberry endophytic bacteria, and silkworm intestinal bacteria against *R. solanacearum*, a mulberry bacterial wilt causative pathogen, were determined using the agar well-diffusion method [32,33]. *R. solanacearum* strain YDqk6 [27] was used for this study, which was stored in the Guangdong Microbial Culture Collection Center (GDMCC NO. 1.1620). Inside a clean workbench, the isolates were inoculated into the prepared nutrient broth (NB) medium. They were then cultured in NB at 28 °C and 180 r/min for 3 days to obtain the strain fermentation broth. Subsequently, holes with a diameter of 5 mm were punched in the indicator medium plates. Then, 50 μL of the strain fermentation broth prepared as described above was added to each hole, while sterile NB medium broth was added to the holes as a control. The plates were placed in an incubator at 28 °C for 24 h. After 24 h, the area around the holes was observed for the presence of an inhibition zone. The isolates that could form an inhibition zone were regarded as biological control bacteria with antibacterial activity, and the diameter of the inhibition zone was measured. All the experiments were conducted in triplicate.

To enable the more widespread application of biological control bacteria in the biological control of common plant diseases, the isolates that exhibited antibacterial activity against *R. solanacearum* were selected, and their antibacterial activities against *Pantoea ananatis* and *Dickeya zeae* were further determined. *P. ananatis* is an unconventional plant pathogen and is also one of the pathogens causing mulberry bacterial wilt [34,35]. *D. zeae* is the major pathogen accountable for maize stalk rot and rice foot rot diseases and has the capacity to infect both monocotyledons and dicotyledons [36]. *P. ananatis* strain LCFJ-001 [35] and *D. zeae* strain EC1 [36] were used in this study. The method for assaying the antimicrobial activity against *P. ananatis* and *D. zeae* was consistent with the assay method for the antimicrobial activity against *R. solanacearum*.

### 2.5. Determination of Plant Growth-Promoting (PGP) Traits of Bacteria

The abilities of bacteria to fix nitrogen, solubilize phosphate, release potassium, and produce siderophores are crucial factors in promoting plant growth. To detect the nitrogen fixation traits of the isolates, they were streaked onto nitrogen-free culture medium plates (Qingdao Hopebiol Biotechnology Co., Ltd., Qingdao, China). A strain that could grow on these plates was considered a nitrogen-fixing bacterium. For the detection of phosphate solubilization traits, 50 μL of each strain’s fermentation broth (prepared as described above) was added to the 5 mm diameter holes in Norganic Phosphorus Bacteria Culture Medium plates (Qingdao Hopebiol Biotechnology Co., Ltd., Qingdao, China). A strain that could form a transparent phosphate-solubilizing zone around the holes was considered a phosphate-solubilizing bacterium, and the diameter of the zone was measured. To detect the potassium release traits of the isolates, they were streaked onto Potassium Bacteria Agar Medium plates (Beijing Coolaber Technology Co., Ltd., Beijing, China). A strain that could grow on these plates was considered a potassium-releasing bacterium. For the detection of siderophore production traits, 50 μL of each strain’s fermentation broth (prepared as described above) was added to a 5 mm diameter hole in CAS Detection Medium plates (Qingdao Hopebiol Biotechnology Co., Ltd., Qingdao, China). A strain that could form an orange halo around the holes was considered a siderophore-producing bacterium, and the diameter of the halo was measured [37]. All the above culture medium plates were incubated at 28 °C for 7 days, and all the treatments were performed in triplicate.

Bacteria can promote plant growth through the synthesis of auxins, with indole-3-acetic acid (IAA) being a prominent example. The IAA production traits of the strains were detected by Salkowski’s colorimetric method [38]. Strains were inoculated into NB medium containing L-tryptophan (100 mg/L) and then incubated in a shaker at 28 °C and 180 r/min for 2 days. After that, the fermentation broth was centrifuged, and the supernatant was mixed with an equal volume of Salkowski’s colorimetric reagent. Then, the mixture was treated in the dark for 30 min. If a red color appeared, it indicated that the strain could synthesize IAA. NB medium without the inoculated strain was mixed with an equal volume of Salkowski’s colorimetric reagent as a negative control, and an IAA standard solution (50 mg/L) was mixed with an equal volume of Salkowski’s colorimetric reagent as a positive control. All the experiments were performed in triplicate.

### 2.6. PCR Detection of Genes Related to Antibiotic Biosynthesis

Biological control bacteria can synthesize secondary metabolites with antibacterial activity, and one of the key biological control mechanisms is the synthesis of antibiotics during strain fermentation, such as surfactin, iturin, and fengycin. To investigate the antibacterial mechanism of biological control bacteria, the present study used the genome of the biological control bacteria as the template. The total DNA of each strain was extracted using the Ezup Column Bacteria Genomic DNA Purification Kit (Sangon Biotechnology Co., Ltd., Shanghai, China), following the manufacturer’s instructions precisely. Using the PCR Instrument T20 (Hangzhou Langji Scientific Instrument Co., Ltd., Hangzhou, China), key genes related to antibiotic synthesis were amplified by PCR technology [21,39,40,41,42]. These genes included those involved in surfactin biosynthesis (*Sfp*, *SfrC*, *SrfAA*), iturin biosynthesis (*ItuA*, *ItuC*, *ItuD*), and fengycin biosynthesis (*FenA*, *FenB*, *FenD*), Bacillomycin biosynthesis (*BAMC*), polyketide synthase (*PKSI*), and nonribosomal peptide synthetase (*NRPS*). The twelve primers used to amplify the functional genes are listed in Appendix A. The PCR amplification products were subjected to 1.2% agarose gel electrophoresis followed by observation using the fully automated gel imaging analysis system ChamGal (Beijing Sage Creation Science Co., Ltd., Beijing, China).

### 2.7. Effects of Plant Growth-Promoting Bacteria on the Growth of Mulberry Saplings

The bacterial strains with good antimicrobial activity or prominent plant growth-promoting traits were selected from the mulberry field bacteria. Each strain was individually inoculated into NB medium and cultured overnight. The overnight-cultured bacteria solution was adjusted to an optical density of OD600 nm = 1 with sterile NB medium. Then, it was centrifuged, and the bacterial inoculum was prepared by resuspending the pellet in 10-fold the volume of the original bacteria solution with sterile water. The same batch of cultivated Guangxi mulberry no. 6 saplings was selected. Their side branches and roots were cut off, leaving only a 6 cm long main root and a 9 cm long main stem, and then they were planted in pots. The pots were randomly divided into eight groups, with 15 plants in each group. For each pot, 60 mL of the single-strain inoculum or an equal-volume mixture of inocula was inoculated into the soil by irrigation. Pots in which 60 mL of sterile water was inoculated served as the control. The inoculations were performed four times (at 0, 15, 30, and 45 days). Sixty days after inoculation, five saplings were randomly selected from each treatment group. The length of the aerial part, and the aerial fresh weight, mulberry leaf yield (leaf fresh weight), root length, root fresh weight (total), and lateral root fresh weight were measured.

### 2.8. Analysis of Bacterial Metabolites Based on the LC-Q/TOF-MS Technique

Metabolites of the plant growth-promoting bacteria were extracted using the methanol extraction method [43]. The target strain was inoculated into NB medium and then cultured on a shaker at 28 °C and 180 r/min for 3 days to obtain the strain fermentation broth. The fermentation broth was centrifuged at 8000 r/min for 10 min, and the supernatant was collected. An equal volume of methanol was added to the supernatant and the mixture was left to extract for 2 h. Then, it was centrifuged at 12,000 r/min for 5 min. The resulting supernatant was filtered through a 0.22 μm microporous filter membrane, placed in the sample vial, and stored in a 4 °C refrigerator. The metabolites of the plant growth-promoting bacteria were analyzed using ultra-high-pressure liquid chromatography quadrupole tandem time-of-flight mass spectrometry (LC-Q/TOF-MS). The metabolites were separated by the chromatographic column eclipse plus C18 (2.1 × 100 mm, 1.8 μm, Agilent), where the mobile phase consisted of A (methanol) and B (0.2% formic acid), and the ESI ion analysis was performed using the Agilent 6540UHD Q-TOF. The data were collected by LC-Q-TOF, and the molecular feature extraction (MFE) was used to identify compounds, with the molecular formula generated based on the CHONSP elements. The presence of lipopeptide antibiotics (surfactin, iturin, fengycin) in the bacterial metabolites was analyzed by fragmentation-based filtering (FBF).

### 2.9. Statistical Analysis

All the data were collected and initially processed using Excel 2016. Statistical analysis was then performed using the Statistical Product and Service Solutions (SPSS 26.0). The bar graphs were drawn using the program GraphPad Prism 8.0.

## 3. Results

### 3.1. Isolation and Composition of the Cultured Bacteria in the Mulberry Field

In this study, 346 bacterial strains were isolated. Among them, 126 were mulberry rhizosphere soil bacteria, accounting for 36.42%; 118 were mulberry endophytic bacteria, accounting for 34.10%; and 102 were silkworm intestinal bacteria, accounting for 29.48% (Figure 1A). Based on the results of the 16S rRNA gene sequencing, the isolated strains were classified into 30 genera (Figure 1B). These genera belonged to Pseudomonadota, Bacillota, and Actinomycetota. The most frequently isolated genera were *Bacillus*, *Pseudomonas*, and *Enterobacter*. *Bacillus* was the most frequently isolated, at a frequency of 25.72%, followed by *Pseudomonas* and *Enterobacter*, both at 20.52% (Figure 1B). Additionally, a large number of *Enterococcus* were isolated, but only from the silkworm intestine; they were not found in the mulberry rhizosphere soil, nor in the roots, stems, and leaves of mulberry.

### 3.2. Diversity of Bacterial Communities in Different Niches in Mulberry Fields

The community structures and genera richnesses of the bacteria varied across different ecological niches in the mulberry fields. Among the isolated mulberry rhizosphere soil bacteria, *Bacillus* accounted for 49.21%, *Pseudomonas* accounted for 19.05%, *Enterobacter* accounted for 7.14%, *Arthrobacter* accounted for 5.56%, *Pantoea* accounted for 3.97%, *Acinetobacter* accounted for 2.38%, and *Brevibacterium*, *Agrobacterium*, and *Serratia* each accounted for 1.59%. Additionally, *Lysinibacillus*, *Bordetella*, *Pseudochrobactrum*, *Luteimonas*, *Stenotrophomonas*, *Alcaligenes*, *Cellulosimicrobium*, *Microbacterium*, *Streptomyces*, and *Staphylococcus* were also isolated (Figure 2A,B). *Bacillus* was the most dominant bacterial genus among the mulberry rhizosphere soil bacteria, followed by *Pseudomonas* and *Enterobacter* (Figure 2A). Among the isolated mulberry endophytic bacteria, *Pseudomonas* accounted for 34.75%, *Enterobacter* accounted for 21.19%, *Bacillus* accounted for 18.64%, *Acinetobacter* accounted for 8.27%, *Pantoea* accounted for 6.78%, and *Stenotrophomonas* accounted for 4.24%. In addition, *Arthrobacter*, *Leucobacter*, *Klebsiella*, *Curtobacterium*, *Erwinia*, *Delftia*, and *Xanthomonas* were also isolated (Figure 2A,C). *Pseudomonas* was the most dominant bacterial genus among the mulberry endophytic bacteria, followed by *Enterobacter* and *Bacillus* (Figure 2A). Among the isolated silkworm intestinal bacteria, *Enterobacter* accounted for 36.27%, *Enterococcus* accounted for 32.35%, *Staphylococcus* accounted for 7.84%, *Pseudomonas* accounted for 5.88%, *Bacillus* accounted for 4.90%, and *Serratia*, *Microbacterium,* and *Micrococcus* each accounted for 1.96%. Additionally, *Stenotrophomonas*, *Klebsiella*, *Aeromonas*, *Brevibacterium Brevundimonas*, *Brachybacterium*, and *Cellulosimicrobium* were also isolated (Figure 2A,D). *Enterobacter* was the most dominant bacterial genus among the silkworm intestinal bacteria, followed by *Enterococcus* and *Staphylococcus* (Figure 2A). *Bacillus*, *Pseudomonas* and *Enterobacter* were isolated in the mulberry rhizosphere soil, mulberry plants, and silkworm intestines. However, their abundances in different niches of the mulberry field varied (Figure 2A). In the mulberry rhizosphere soil bacteria, the relative abundance order of the bacteria was *Bacillus* > *Pseudomonas* > *Enterobacter*. In the mulberry endophytic bacteria, the relative abundance order of the bacteria was *Pseudomonas* > *Enterobacter* > *Bacillus*. In the silkworm intestinal bacteria, the relative abundance order of the bacteria was *Enterobacter* > *Pseudomonas* > *Bacillus*. *Bacillus*, *Pseudomonas* and *Enterobacter* were not only the predominant genera among the mulberry rhizosphere soil bacteria communities but also among the mulberry endophytic bacteria communities. However, the predominant genera among the silkworm intestinal bacteria communities were *Enterobacter*, *Enterococcus*, and *Staphylococcus*.

### 3.3. Plant Growth-Promoting Bacterial Composition and Diversity Analysis

Among the 346 bacterial strains, 30 bacterial strains (accounting for 1.96%) exhibited antimicrobial activity against *R. solanacearum*, 92 bacterial strains (accounting for 26.59%) exhibited nitrogen fixation activity, 209 bacterial strains (accounting for 60.40%) exhibited phosphate solubilization activity, 197 bacterial strains (accounting for 56.93%) exhibited potassium release activity, 179 bacterial strains (accounting for 51.73%) exhibited siderophore production activity, and 308 bacterial strains (89.02% of the total) had at least one of the above activities (Figure 3B). The 308 bacterial strains with nitrogen fixation, phosphate solubilization, potassium release, siderophore production or antibacterial activity were composed of multiple genera (Figure 3A). *Bacillus* was the predominant genus among the biological control bacteria. In contrast, *Enterobacter* demonstrated remarkable advantages in terms of nitrogen fixation, phosphate solubilization, potassium release or siderophore production.

### 3.4. Screening of Plant Growth-Promoting Bacteria

#### 3.4.1. Determination of Antibacterial Activity and Antibacterial Spectrum of Biological Control Bacteria

Among the 30 biological control bacteria against *R. solanacearum*, *Bacillus* was predominant, accounting for 60.00% (18 of 30 strains), followed by *Pseudomonas* (26.67%, 8 of 30 strains), with 2 strains of *Enterobacter*, 1 strain of *Serratia*, and 1 strain of *Lysinibacillus*. In terms of the antibacterial activity against *R. solanacearum*, 88.89% (16 of 18) of the *Bacillus* strains had an inhibition zone diameter of 10 mm or more, while for *Pseudomonas*, the proportion was 62.50% (5 of 8), and the diameters for *Enterobacter*, *Serratia*, and *Lysinibacillus* were all less than 10 mm. Regarding the antibacterial activity against *P. ananatis*, 66.67% (20 of 30) of the strains showed an antibacterial effect, mainly from *Bacillus* (83.33%, 15 of 18) and *Pseudomonas* (62.50%, 5 of 8). As for the antibacterial activity against *D. zeae*, 73.33% (22 of 30) of the strains had an antibacterial effect, predominantly from *Bacillus* (88.89%, 16 of 18) and *Pseudomonas* (75.00%, 6 of 8). Among these bacteria, some specific strains showed distinct antibacterial performance. For example, both *Bacillus* sp. strain TF-2 and *Bacillus* sp. strain NY-1 had an inhibition zone diameter of 10 mm or more against *R. solanacearum*. *Bacillus* sp. strain TF-2 had the strongest antibacterial activity against *R. solanacearum*. However, *Bacillus* sp. strain TF-2 did not show antibacterial activity against *P. ananatis* and *D. zeae*. *Bacillus* sp. strain NY-1 showed stable and strong antibacterial activity against *R. solanacearum*, *P. ananatis*, and *D. zeae* (Table 1).

#### 3.4.2. Analysis of the Plant Growth-Promoting Properties of Biological Control Bacteria

Among the 30 biological control bacterial strains against *R. solanacearum*, only 4 bacterial strains (accounting for 11.11%) possessed nitrogen fixation activity, and the other 26 bacterial strains barely grew on nitrogen-free culture medium, 16 bacterial strains (accounting for 53.33%) showed phosphate solubilization activity, 19 bacterial strains (accounting for 63.33%) showed potassium release activity, 17 bacterial strains (accounting for 56.67%) showed siderophore production activity, and 9 bacterial strains (accounting for 30.00%) showed IAA production activity. Although *Bacillus* sp. strain NY-1 previously showed stable and strong antibacterial activity against *R. solanacearum*, *P. ananatis* and *D. zeae*, it did not have the activity of nitrogen fixation, phosphate solubilization, siderophore production and IAA production (Table 2).

#### 3.4.3. Analysis of Antibacterial Mechanism of Biological Control Bacteria

The preliminary analysis results concerning the antimicrobial mechanism showed that at least one antibiotic biosynthesis-related gene could be amplified in all 30 target strains, and the gene *NRPS* was positively amplified in all the strains. Secondly, the detection rate of the gene *PKSI* was the highest, accounting for 80.00% (24 of 30 strains). All the *Bacillus* strains could amplify at least one lipopeptide antibiotic biosynthesis-related gene, and no lipopeptide antibiotic biosynthesis-related gene was amplified in any non-*Bacillus* strains. Among all 18 biological control *Bacillus* strains, the gene *SrfC* had the highest detection rate at 66.7% (12 of 18 strains), followed by the gene *Sfp* with a detection rate of 61.11% (11 of 18 strains), both of which were related to surfactin biosynthesis. *SrfAA*, another gene associated with surfactin biosynthesis, was detected at 33.33% (6 of 18 strains). *ItuA* and *ItuD*, genes involved in iturin biosynthesis, as well as *FenA* and *FenB*, genes involved in fengycin biosynthesis, were also detected. However, the genes *ItuC* and *FenD* were not detected. *Bacillus* sp. strain TF-2 tested positive for the genes *SrfC* and *ItuD*, and *Bacillus* sp. strain NY-1 tested positive for the genes *Sfp*, *SrfC* and *ItuD* (Figure 4).

#### 3.4.4. Determination and Analysis of Plant Growth-Promoting Properties

The results concerning the plant growth-promoting properties showed that 58 bacterial strains had both nitrogen fixation, phosphate solubilization, potassium release and siderophore production activity. *Enterobacter* sp. strain JCS-3 had the strongest activity of phosphate solubilization (the phosphate-solubilizing zone had the largest diameter) and outstanding nitrogen fixation, potassium release, siderophore production properties, and it also had the property of IAA production (Figure 5).

The 16S rRNA gene amplification product was ligated to the pBM23 vector, followed by sequencing analysis to identify the bacterial species. Specifically, TF-2 was identified as *Bacillus licheniformis*, NY-1 was identified as *Bacillus subtilis*, and JCS-3 was identified as *Enterobacter hormaechei* (Table 3). Subsequently, the obtained sequences were submitted to the NCBI GenBank database, and the accession numbers were obtained.

### 3.5. Evaluation of Mulberry Sapling Growth

*Bacillus licheniformis* strain TF-2 was isolated from the mulberry rhizosphere soil bacteria, *Bacillus subtilis* strain NY-1 from the mulberry endophytic bacteria, and *Enterobacter hormaechei* strain JCS-3 from the silkworm intestinal bacteria. All these strains exhibited outstanding antibacterial activity or plant growth-promoting properties. Thus, they were selected as single or mixed inoculants for application to mulberry, and their effects on mulberry growth were further evaluated (Figure 6). The results showed that all the plant growth-promoting bacterial inoculants stimulated the growth of mulberry saplings to different levels compared to the water-treated control. For the single inoculants, *B. licheniformis* strain TF-2 significantly increased the root fresh weight (including the total root fresh weight and lateral root fresh weight) with a significant difference (*p* < 0.05). *B. subtilis* strain NY-1 led to a significant increase in the total root fresh weight of the saplings. When treated with *E. hormaechei* strain JCS-3, all the measured growth parameters of the saplings increased significantly, except for the length of the aerial part. For the mixed inoculants, only in the TF-2 + NY-1 treatment, the aerial part length and root length did not change significantly, while the growth parameters of the other mixed inoculants treatments all increased significantly. The equal volume mixed inoculants of the three bacterial inoculum showed the highest PGP activities. Specifically, the aerial part length, aerial fresh weight and leaf fresh weight increased by 20.82%, 36.63% and 86.55%, respectively. The root length, total root fresh weight and lateral root fresh weight increased by 34.02%, 52.20% and 96.98%, respectively.

### 3.6. Putative Bioactive Compounds Detected by LC-Q/TOF-MS Analysis

Surfactin, iturin, and fengycin were not detected in the metabolite of *B. licheniformis* strain TF-2. In the metabolite of *B. subtilis* strain NY-1, iturin and fengycin were not detected, but surfactin (C_53_H_93_N_7_O_13_) was detected (Figure 7). The target mass (reference mass) of the target formula C_53_H_93_N_7_O_13_ detected by ESI was 1035.6831. When electron spray ionization (ESI) in negative (NEG) ion mode was used to detect the metabolite C_53_H_93_N_7_O_13_ of *B. subtilis* strain NY-1, the observation mass was 1035.6803, the target mass error was −2.76 ppm, the observation mass to charge ratio (*m*/*z*) was 1034.6732, the retention time was 11.063 min, and the abundance was 87,789 (Figure 7A,C,E). When ESI in positive (POS) ion mode was used to detect the metabolite C_53_H_93_N_7_O_13_ of *B. subtilis* strain NY-1, the observation mass was 1035.6843, the target mass error was 1.16 ppm, the observation mass-to-charge ratio (*m*/*z*) was 1036.6918, the retention time was 11.067 min, and the abundance was 79,774 (Figure 7B,D,F).

## 4. Discussion

Soil bacteria have the ability to enhance the bioavailability of soil nutrients, thereby promoting plant growth [44]. Additionally, they can indirectly contribute to plant growth by inhibiting plant pathogens [45]. Consequently, they are widely recognized as plant growth-promoting bacteria (PGPB). The interaction between plant rhizosphere soil bacteria and plants is extremely intimate [46]. Plant endophytic bacteria, similar to plant rhizosphere soil bacteria, are capable of promoting plant growth. Moreover, they can even interact with plants more efficiently than plant rhizosphere soil bacteria [47,48,49]. Animal feces, which was once used as an important fertilizer, will eventually return to the soil. However, notably, no previous research has included intestinal bacteria within the scope of PGPB. In this study, taking the safety of silkworms into account, the intestinal bacteria of silkworms were incorporated into the research scope of PGPB. This is because the safety of PGPB for silkworms is a crucial factor when applying them in mulberry fields, and the bacteria isolated from healthy silkworms are considered to pose no risk to silkworms.

In this study, some bacteria could be simultaneously present in the mulberry rhizosphere soil, mulberry and silkworm intestines, but their abundance was not the same, and the relative abundance of the bacteria would change according to the ecological niche environment (Figure 2). For example, in terms of *Bacillus* and *Pseudomonas*, we isolated a large number of them in the mulberry rhizosphere soil and mulberry, but rarely in the intestines of silkworms. *Enterobacter* could be isolated in the rhizosphere soil of mulberry, mulberry and silkworm intestines, but there was a trend of increasing bacteria in the silkworm intestines. We are curious about which bacteria will flow in this cycle and how the microflora changes during this flow, and we need to conduct more research.

Currently, culture-dependent methods and culture-independent methods have been widely used to analyze bacterial communities in different environments [27,50]. Especially, the development and application of next-generation sequencing (NGS) technology, as well as culture-independent methods based on 16S rDNA amplicon sequencing in the Illumina MiSeq and HiSeq systems, have played an important role in the analysis of complex microbial communities, but culture-dependent methods are still essential. Most information on microbial diversity was obtained using conventional culture techniques. Moreover, the development and utilization of bacteria, the excavation of engineering strains, and the genetic engineering technology of bacteria are all based on the culturable bacteria. In this study, 346 cultivable bacterial strains of 30 genera were isolated from mulberry rhizosphere soil, mulberry and silkworm intestines, derived from *Pseudomonadota*, *Bacillota* and *Actinobacteria*. However, culture-dependent methods are also limited to some extent. The bacteria isolated by culture-dependent methods are basically from *Pseudomonadota*, *Bacillota*, *Actinobacteria* and *Bacteroidota* [21,27]. Culture-independent methods often allow us to find some bacteria that cannot be isolated by culture-dependent methods, such as *Acidobacteria*, *Armatimonadetes*, *Chlamydiae*, *Chlorobi*, etc. [27]. The reason that some bacteria found in culture-independent methods are not isolated by culture-dependent methods may be that on the one hand, the relative abundance of these bacteria is too low, while on the other hand, it may be limited by the culture conditions, given that we cannot completely simulate the natural growth environment of bacteria. To simulate the natural growth conditions of bacteria as much as possible, or to find some methods suitable for cultivating more bacteria, so that we can study bacteria more deeply, is also a problem to be broken through.

PGPB can promote plant growth through either direct or indirect means [9,11,49]. When bacteria facilitate plants’ access to nutrients or participate in the regulation of plant hormones, it is considered a direct promotion of plant growth. When PGPB inhibit plant pathogens or enhance plant resistance, they are considered to promote plant growth indirectly. In this study, *Bacillus licheniformis* strain TF-2, *Bacillus subtilis* strain NY-1, and *Enterobacter hormaechei* strain JCS-3 showed direct or indirect promotion effects on plant growth. *E. hormaechei* strain JCS-3 exhibited strong activities in terms of nitrogen fixation, phosphate solubilization, potassium release, siderophore production, and IAA production. It could promote plants’ acquisition of nitrogen, phosphorus, potassium, iron, and other nutrients, thus directly promoting plant growth (Figure 5 and Figure 6). *B. licheniformis* strain TF-2 and *B. subtilis* strain NY-1 were less active in promoting plant nutrient uptake and could not produce IAA. They mainly promoted plant growth indirectly by inhibiting plant pathogens (Table 1 and Table 2 and Figure 6). When saplings were inoculated with a single agent, *E. hormaechei* strain JCS-3 was found to have more pronounced plant growth-promoting capacity than the other two strains. This might imply that direct plant growth promotion by PGPB is more critical than indirect promotion. However, when inoculated with a mixture, we found that the mixture had better plant growth-promoting ability than inoculating the saplings with either of the bacteria alone. This was especially true for the mixture of bacteria that directly promote plant growth and those that indirectly promote plant growth. *Bacillus* sp. strain CW16-5 showed strong plant growth promotion potential [21]. When plants were inoculated with this strain, the shoot length increased by up to 83.37% and the root fresh weight increased by 217.70%, respectively. This significant growth-promoting effect might be attributed to its excellent antibacterial and phosphate-solubilizing activities. However, inoculants consisting of single bacteria often struggle to meet the complex nutrient requirements of plants. The *Bacillus* sp. strain CW16-5 had multiple growth-promoting properties for plants but failed to produce siderophores [21]. *B. subtilis* strain KJ-2 and *B. amyloliquefaciens* strain WK-2 had multiple growth-promoting properties for plants, including siderophore production, but failed to produce IAA [33]. Inoculants with a mixture of multiple bacteria can satisfy the plant demand for complex nutrients to a greater extent.

The inhibition of one bacterium by another may be achieved through the production of antibiotic substances [51,52]. Before testing the metabolites of the strains, we amplified the key genes of the biocontrol bacteria by PCR. Theoretically, the test results concerning the key genes involved in antibacterial synthesis should directly correspond to the detection results of the metabolites. However, the experimental results show that this is not always the case. This may be due to the presence of the key gene involved in antimicrobial synthesis, but because it is limited by some factors, the key gene involved in antimicrobial synthesis is not expressed, so the key gene for antimicrobial synthesis is detected on the strain and the corresponding metabolites are not detected. In the detection of key genes related to antimicrobial synthesis in *B. licheniformis* strain TF-2, the genes *SrfC* and *ItuD* tested positive. However, the presence of surfactin and iturin was not detected in its metabolites. This implies that *B. licheniformis* strain TF-2 may possess some other antimicrobial substances in addition to surfactin, iturin, and fengycin. Compound information on the unknown metabolites of *B. licheniformis* strain TF-2 was extracted by molecular feature extraction (MFE) technique. 6-gingerol (C_17_H_26_O_4_) and 3H-1,2-dithiole-3-thione (C_3_H_2_S_3_) were extracted from the metabolites of *B. licheniformis* strain TF-2 in electron spray ionization (ESI) negative (NEG) ion mode (Figure 8). It has been reported that 6-gingerol exhibits potent anti-mycobacterial activity both in vitro and in vivo [53]. Moreover, 3H-1,2-dithiole-3-thione is also a potential compound with antibacterial activity [54]. 6-gingerol (C_17_H_26_O_4_) and 3H-1,2-dithiole-3-thione (C_3_H_2_S_3_) might be potential antibacterial metabolites of *B. licheniformis* strain TF-2. For 6-gingerol, the target mass (reference mass) was 294.1834, while the observation mass and the observation mass-to-charge ratio (m/z) were both 293.1761, and the retention time was 6.86 min (Figure 8A,C,E). For 3H-1,2-dithiole-3-thione, the target mass (reference mass) was 133.9183, while the observation mass and the observation mass-to-charge ratio (m/z) were both 132.9233, and the retention time was 10.935 min (Figure 8A,D,F). For *B. subtilis* NY-1, the genes *Sfp*, *SrfC* and *ItuD* tested positive, but only the presence of surfactin was detected in its metabolites. Surfactin is a cyclic lipopeptide biosurfactant composed of four isomers (surfactin A–D) that can be used as an antimicrobial adjuvant, with antibacterial, antifungal, mycoplasma resistant and hemolytic effects, and it also has antiviral activity against a variety of enveloped viruses [55,56,57,58]. The most important representatives of the iturin family are iturin A, C, D and E, and variants of mycosubtilin and bacillomycin, and the fengycin family is represented by many isoforms of fengycin, plipastatin and maltacin [59,60,61]. In addition, the presence but absence of expression of the genes may also lead to the corresponding metabolites not being detected. Further studies are needed on the antibacterial mechanism of biocontrol bacteria.

## 5. Conclusions

The present study showed that PGPB were abundant in mulberry fields. PGPB were widely found in different ecological niches of mulberry fields, including the mulberry rhizosphere soil, the mulberry, and the silkworm intestine. The intestinal bacteria of silkworms had strong activities in terms of nitrogen fixation, phosphate solubilization, potassium release and siderophore production. Among the 346 bacterial isolates, 30 bacterial strains showed antibacterial activity against *R. solanacearum*, and 58 bacterial strains had nitrogen fixation, phosphate solubilization, potassium release and siderophore production properties. *B. licheniformis* strain TF-2 showed the most antibacterial activity against *R. solanacearum*. *B. subtilis* strain NY-1 not only had antibacterial activity against *R. solanacearum* but also against *P. ananatis* and *D. zeae*. *E. hormaechei* strain JCS-3 had the most phosphate solubilization activity and also had outstanding nitrogen fixation, potassium release, siderophore production and IAA production. The results showed that these three bacterial strains had some growth-promoting effect on the mulberry seedlings, and the combination of the three bacterial strains promoted the growth of mulberry seedlings more strongly than a single inoculum. Our findings indicated that the silkworm intestine is a niche rich in PGPB and that silkworm intestinal bacteria may be an important source of PGPB.

## Figures and Tables

**Figure 1 microorganisms-13-01012-f001:**
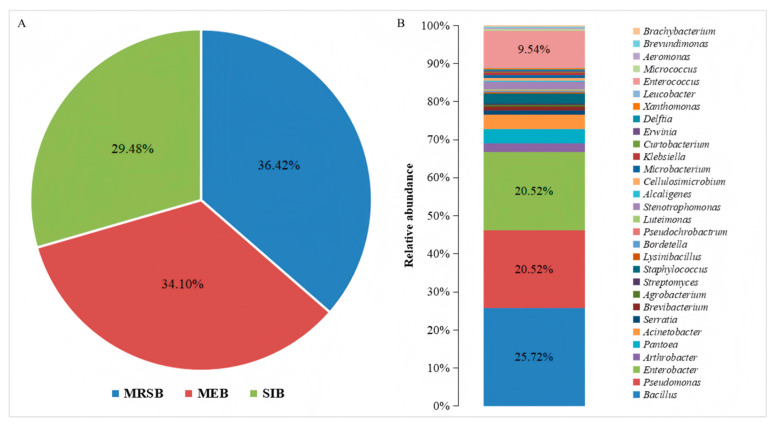
Bacterial composition and distribution of bacterial genera isolated from different niches in mulberry fields. (**A**) The composition of culturable bacteria isolated from different niches in the mulberry fields, where MRSB, MEB, and SIB represent the communities of mulberry rhizosphere soil bacteria, mulberry endophytic bacteria, and silkworm intestinal bacteria, respectively. (**B**) The distribution of the genera of the culturable bacteria isolated from the mulberry fields.

**Figure 2 microorganisms-13-01012-f002:**
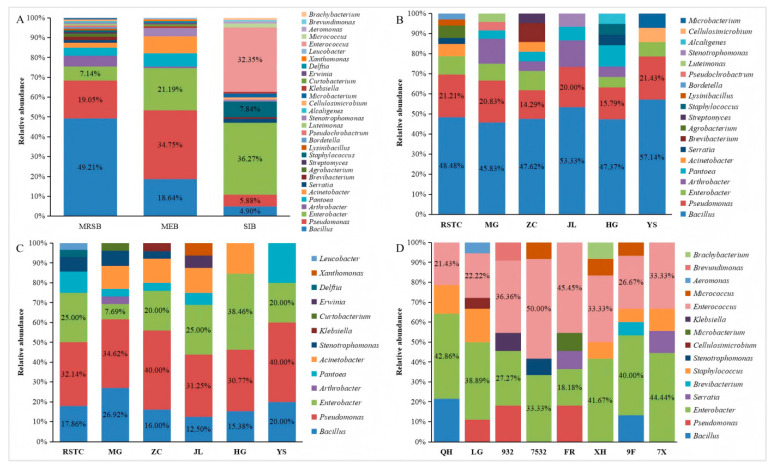
Relative abundances (%) of cultivable bacteria isolated from different niches in mulberry fields. (**A**) Communities isolated from different niches in the mulberry fields, where MRSB, MEB, and SIB represent the communities of mulberry rhizosphere soil bacteria, mulberry endophytic bacteria, and silkworm intestinal bacteria, respectively. (**B**) and (**C**) Communities of mulberry rhizosphere soil bacteria and mulberry endophytic bacteria isolated from different locations, where RSTC, MG, ZC, JL, HG, and YS represent the communities isolated from the Regional Sericulture Training Center for Asia-Pacific, South China Agricultural University, Zengcheng District, Jiulong Town, Hanguang Town, and Yangshan County, respectively. (**D**) Communities isolated from different silkworm varieties, QH, LG, 932, 7532, FR, XH, 9F, and 7X represent the communities isolated from Qiu Hua × Ping 30, Liang Guang No. 29327532, Furong, Xianghui, 932 × Furong, and 7532 × Xianghui, respectively.

**Figure 3 microorganisms-13-01012-f003:**
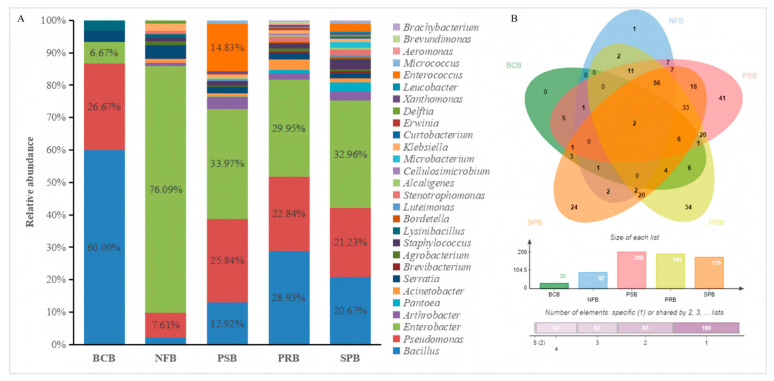
Analyses of the Venn diagram (**A**) and genera distribution (**B**) for 308 bacteria exhibiting antibacterial activity or plant growth-promoting activity. BCB, NFB, PSB, PRB and SPB represent the biological control bacteria, nitrogen-fixing bacteria, phosphate-solubilizing bacteria, potassium-releasing bacteria, and siderophore-producing bacteria, respectively.

**Figure 4 microorganisms-13-01012-f004:**
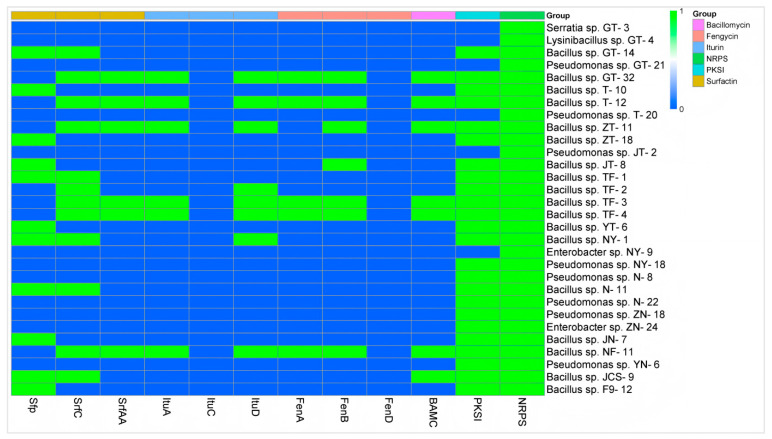
Determination of the genes associated with antibiotic biosynthesis in 30 biological control bacteria. The target genes were amplified by PCR, with positive results represented by 1 and negative results represented by 0.

**Figure 5 microorganisms-13-01012-f005:**
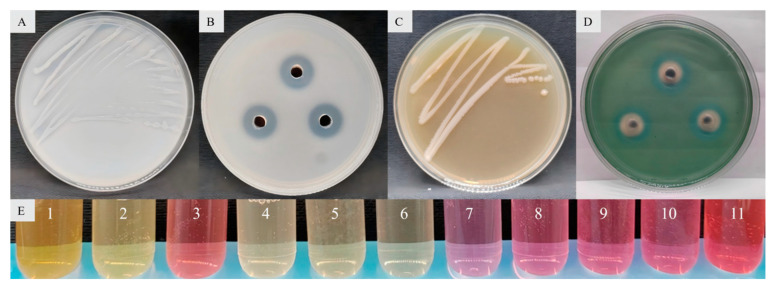
Properties of nitrogen fixation (**A**), phosphate solubilization (**B**), potassium release (**C**), siderophore production (**D**), and IAA production (**E**) in *Enterobacter* sp. strain JCS-3, where 1 represents *Bacillus* sp. strain TF-2, 2 represents the *Bacillus* sp. strain NY-1, 3 represents the *Enterobacter* sp. strain JCS-3, 4 and 5 represent two other tested strains, respectively, 6 represents the negative control, and 7 to 11 represent the positive controls with IAA contents of 10 mg/L, 20 mg/L, 30 mg/L, 40 mg/L, and 50 mg/L, respectively.

**Figure 6 microorganisms-13-01012-f006:**
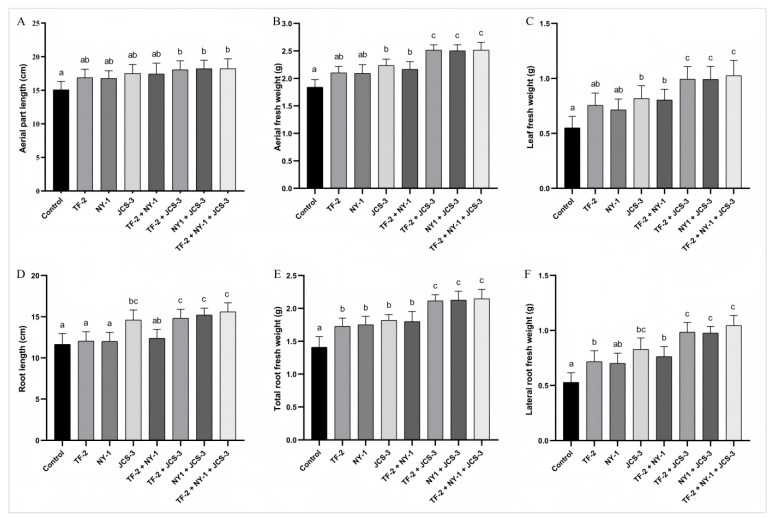
Effects of the plant growth-promoting bacterial strains on the growth of mulberry saplings. (**A**) aerial part length; (**B**) aerial fresh weight; (**C**) leaf fresh weight; (**D**) root length; (**E**) total root weight; and (**F**) lateral root fresh weight. The data are presented as the mean ± SD of five replicates. Different letters on the bars represent treatment groups that show significant differences at the *p* < 0.05 confidence level by one-way analysis of variance (ANOVA) and the least significant difference (LSD) test.

**Figure 7 microorganisms-13-01012-f007:**
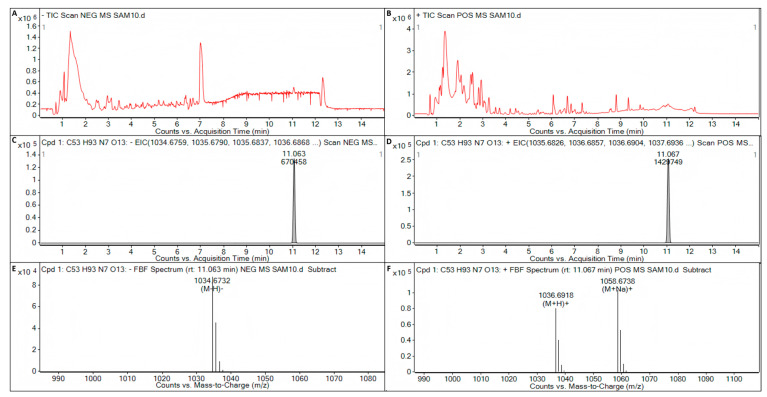
LC-Q/TOF-MS analysis of the metabolite of *B. subtilis* strain NY-1. (**A**) Total ion chromatogram (TIC) of the electron spray ionization (ESI) for the metabolites of *B. subtilis* strain NY-1 in negative (NEG) ion mode. (**B**) TIC of the ESI for the metabolites of *B. subtilis* strain NY-1 in positive (POS) ion mode. (**C**) Extracted ion chromatogram (EIC) of the ESI for the metabolite C_53_H_93_N_7_O_13_ of *B. subtilis* strain NY-1 in NEG ion mode. (**D**) EIC of the ESI for the metabolite C_53_H_93_N_7_O_13_ of *B. subtilis* strain NY-1 in POS ion mode. (**E**) Mass spectrum (MS) of the ESI for the metabolite C_53_H_93_N_7_O_13_ of *B. subtilis* strain NY-1 in NEG ion mode. (**F**) MS of the ESI for the metabolite C_53_H_93_N_7_O_13_ of *B. subtilis* strain NY-1 in POS ion mode.

**Figure 8 microorganisms-13-01012-f008:**
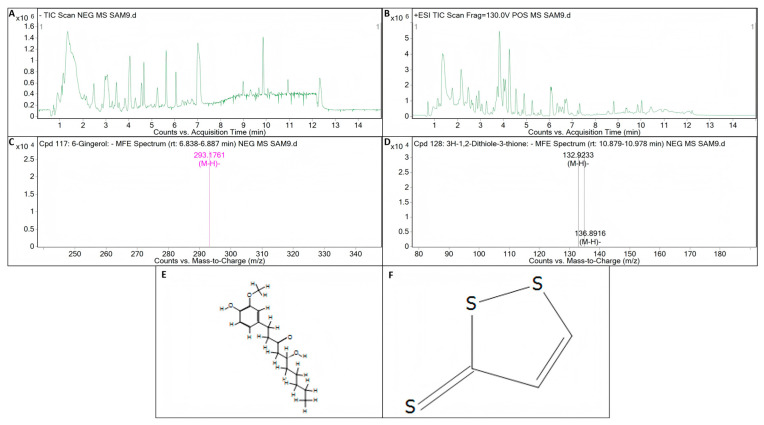
LC-Q/TOF-MS analysis of the metabolite of *B. licheniformis* strain TF-2. (**A**) Total ion chromatogram (TIC) of the electron spray ionization (ESI) for the metabolites of *B. licheniformis* strain TF-2 in negative (NEG) ion mode. (**B**) TIC of the ESI for the metabolites of *B. licheniformis* strain TF-2 in positive (POS) ion mode. (**C**) Mass spectrum (MS) of the ESI for the metabolite C_17_H_26_O_4_ of *B. licheniformis* strain TF-2 in NEG ion mode. (**D**) MS of the ESI for the metabolite C_3_H_2_S_3_ of *B. licheniformis* strain TF-2 in NFG ion mode. (**E**) Compound structure of 6-gingerol (C_17_H_26_O_4_). (**F**) Compound structure of 3H-1,2-dithiole-3-thione (C_3_H_2_S_3_).

**Table 1 microorganisms-13-01012-t001:** Determination of the antibacterial activity and antibacterial spectrum of 30 biological control bacteria against *Ralstonia solanacearum*.

No.	Strain	Inhibition Zone		
		*R. solanacearum*	*P. ananatis*	*D. zeae*
1	*Serratia* sp. GT-3	9.33 ± 0.58	-	-
2	*Lysinibacillus* sp. GT-4	6.33 ± 0.58	-	-
3	*Bacillus* sp. GT-14	13.00 ± 0.00	8.67 ± 0.58	11.00 ± 0.00
4	*Pseudomonas* sp. GT-21	8.67 ± 0.58	-	-
5	*Bacillus* sp. GT-32	14.00 ± 1.00	10.67 ± 0.58	11.33 ± 0.58
6	*Bacillus* sp. T-10	10.00 ± 1.00	8.33 ± 0.58	9.67 ± 0.58
7	*Bacillus* sp. T-12	13.33 ± 0.58	10.00 ± 1.00	11.00 ± 0.00
8	*Pseudomonas* sp. T-20	10.67 ± 0.58	8.00 ± 0.00	9.00 ± 0.00
9	*Bacillus* sp. ZT-11	14.67 ± 0.58	12.00 ± 1.00	12.67 ± 0.58
10	*Bacillus* sp. ZT-18	9.00 ± 0.00	-	8.00 ± 0.00
11	*Pseudomonas* sp. JT-2	9.00 ± 1.00	-	-
12	*Bacillus* sp. JT-8	10.67 ± 0.58	7.33 ± 0.58	10.00 ± 0.00
13	*Bacillus* sp. TF-1	14.33 ± 0.58	11.67 ± 0.58	12.33 ± 1.15
14	*Bacillus* sp. TF-2	15.67 ± 0.58	-	-
15	*Bacillus* sp. TF-3	11.00 ± 0.00	8.33 ± 0.58	10.67 ± 0.58
16	*Bacillus* sp. TF-4	14.33 ± 1.15	11.33 ± 0.58	12.00 ± 1.00
17	*Bacillus* sp. YT-6	11.00 ± 0.00	9.00 ± 0.00	10.00 ± 0.00
18	*Bacillus* sp. NY-1	15.00 ± 0.00	13.67 ± 1.15	15.33 ± 0.58
19	*Enterobacter* sp. NY-9	8.67 ± 0.58	-	-
20	*Pseudomonas* sp. NY-18	13.00 ± 0.00	10.67 ± 0.58	12.00 ± 1.00
21	*Pseudomonas* sp. N-8	9.33 ± 0.58	-	9.00 ± 0.00
22	*Bacillus* sp. N-11	13.33 ± 1.15	10.33 ± 0.58	12.33 ± 0.58
23	*Pseudomonas* sp. N-22	13.33 ± 0.58	10.00 ± 1.00	12.67 ± 0.58
24	*Pseudomonas* sp. ZN-18	12.67 ± 1.15	9.00 ± 0.00	12.00 ± 0.00
25	*Enterobacter* sp. ZN-24	7.33 ± 0.58	-	-
26	*Bacillus* sp. JN-7	12.00 ± 1.00	9.67 ± 0.58	11.67 ± 0.58
27	*Bacillus* sp. NF-11	13.33 ± 0.58	10.67 ± 0.58	12.33 ± 1.15
28	*Pseudomonas* sp. YN-6	11.33 ± 0.58	8.67 ± 0.58	11.00 ± 0.00
29	*Bacillus* sp. JCS-9	10.00 ± 0.00	7.33 ± 0.58	8.67 ± 0.58
30	*Bacillus* sp. F9-12	8.33 ± 1.15	-	-

**Table 2 microorganisms-13-01012-t002:** Determination of the PGP properties of 30 biological control bacteria against *R. solanacearum.*

No.	Strain	PGP Properties				
		NF	PB	PR	SP	IAA
1	*Serratia* sp. GT-3	+	9.67 ± 1.15	+	12.33 ± 0.58	+
2	*Lysinibacillus* sp. GT-4	+	−	−	7.00 ± 0.00	+
3	*Bacillus* sp. GT-14	−	−	−	8.67 ± 0.58	−
4	*Pseudomonas* sp. GT-21	−	−	−	14.33 ± 1.15	−
5	*Bacillus* sp. GT-32	−	−	+	−	−
6	*Bacillus* sp. T-10	−	7.67 ± 0.58	−	−	−
7	*Bacillus* sp. T-12	−	−	+	8.00 ± 0.00	−
8	*Pseudomonas* sp. T-20	−	8.33 ± 1.15	+	11.00 ± 0.00	+
9	*Bacillus* sp. ZT-11	−	7.33 ± 0.58	+	−	−
10	*Bacillus* sp. ZT-18	−	7.00 ± 0.00	−	−	−
11	*Pseudomonas* sp. JT-2	−	8.00 ± 1.00	+	12.00 ± 0.00	+
12	*Bacillus* sp. JT-8	−	6.67 ± 0.58	+	8.33 ± 0.58	−
13	*Bacillus* sp. TF-1	−	−	+	−	−
14	*Bacillus* sp. TF-2	−	7.33 ± 0.58	**+**	7.33 ± 0.58	**−**
15	*Bacillus* sp. TF-3	−	−	+	−	−
16	*Bacillus* sp. TF-4	−	−	+	−	+
17	*Bacillus* sp. YT-6	−	−	+	−	+
18	*Bacillus* sp. NY-1	−	−	**+**	−	**−**
19	*Enterobacter* sp. NY-9	+	11.67 ± 1.15	+	12.00 ± 1.00	+
20	*Pseudomonas* sp. NY-18	−	11.33 ± 0.58	−	−	−
21	*Pseudomonas* sp. N-8	−	9.67 ± 1.15	−	−	−
22	*Bacillus* sp. N-11	−	7.33 ± 0.58	−	−	−
23	*Pseudomonas* sp. N-22	−	8.33 ± 0.58	+	9.33 ± 1.15	−
24	*Pseudomonas* sp. ZN-18	−	8.00 ± 1.00	−	12.00 ± 1.00	−
25	*Enterobacter* sp. ZN-24	+	10.67 ± 0.58	−	−	+
26	*Bacillus* sp. JN-7	−	−	+	7.00 ± 0.00	−
27	*Bacillus* sp. NF-11	−	−	+	8.33 ± 0.58	−
28	*Pseudomonas* sp. YN-6	−	8.33 ± 0.58	+	9.67 ± 1.15	−
29	*Bacillus* sp. JCS-9	−	−	+	8.67 ± 0.58	−
30	*Bacillus* sp. F9-12	−	−	−	10.33 ± 0.58	+

NF, PB, PR, SP and IAA represent nitrogen fixation, phosphate solubilization, potassium release, siderophore production, and indole-3-acetic acid (IAA) production, respectively. “+” represents positive, and “−” represents negative.

**Table 3 microorganisms-13-01012-t003:** BLASTn results of the 16S rRNA sequences.

Isolate	Best NCBI Database Match with Accession Number	% Identity	GenBank Accession Number of Submitted Sequence
TF-2	*Bacillus licheniformis* strain CO-12 (OL354425.1)	99.54%	PP596549.1
NY-1	*Bacillus subtilis* strain K21 (JN587510.1)	99.74%	MT669326
JCS-3	*Enterobacter hormaechei* strain C44 (CP042566.1)	99.67%	PP440228.1

## Data Availability

The original contributions presented in this study are included in the article/Appendix A. Further inquiries can be directed to the corresponding author.

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
