# Peer review of "Analysis of the Changes in Diversity of Culturable Bacteria in Different Niches of Mulberry Fields and Assessment of Their Plant Growth-Promoting Potential"

_microorganisms, 2025, doi:10.3390/microorganisms13051012_

Round 1
Reviewer 1 Report
Comments and Suggestions for Authors
Dear Authors,
Well done for a very interesting and comprehensive research manuscript. The study seem to have been appropriately designed and experiments executed in an acceptable manner. Just a few suggestions and comments:
- The introduction needs to explain more clearly and adequately why it was interesting to investigate the connections between mulberry rhizosphere soil bacteria, mulberry endophytic bacteria, and silkworm intestinal bacteria. It may require clarity that the silkworms feed on actively growing mulberry plants and not harvested and processed mulberry leaves.
- Lines 50-53 does not quite add up, please rearrange or explain better. its not clear how soil compaction is connected to the use of chemical fertilizers and bactericides.
- Line 141- what is an appropriate colony density
- Line 161-165. This can be improved or summarised so that it may read better e.g The antimicrobial activity of mulberry rhizosphere soil bacteria,mulberry endophytic bacteria, and silkworm intestinal bacteria against R. solanacearum, a mulberry bacterial wilt causative pathogen, were determined using the agar well-diffusion method.
- Line 378- It is not surprising that only 11.1% bacterial strains were capable of nitrogen fixation. Nitrogen fixing babteria tends to rely mainly on symbiotic relationships in order to promote plant growth by mechanisms such as nitrogen fixation and antimicrobial properties against other competing microbes in the soil
- I do appreciate that the authors tried to expand their study as much as possible and produced a significant amount of data, however it would be interesting if a follow up study is done using next generation sequencing technology to identify microbials in the intestines of silkworms that can not be cultured.
- I think Line 494-502 is more appropriate in the introduction
The quality of English is acceptable in my opinion. A few grammatical errors may require some attention.
Author Response
Dear Reviewer,
Thank you very much for your valuable comments on our manuscript. Your feedback plays a very important role in improving the quality of manuscript and improving the research content. In accordance with your comments, we have made a comprehensive and detailed revision to the manuscript, and now we would like to report the details of the revisions to you point by point.
Comments 1: The introduction needs to explain more clearly and adequately why it was interesting to investigate the connections between mulberry rhizosphere soil bacteria, mulberry endophytic bacteria, and silkworm intestinal bacteria. It may require clarity that the silkworms feed on actively growing mulberry plants and not harvested and processed mulberry leaves.
Response 1: Thank you for pointing this out. We agree with this comment. Therefore, in the introduction, we have explained more clearly and adequately why it is interesting to investigate the connections between mulberry rhizosphere soil bacteria, mulberry endophytic bacteria, and silkworm intestinal bacteria. This change
can be found in lines 70-84 of the revised manuscript. In addition, we have made it clear that the silkworms feed on actively growing mulberry plants and not harvested and processed mulberry leaves. This change can be found in lines 48-49 of the revised manuscript.
Comments 2: Lines 50-53 does not quite add up, please rearrange or explain better. its not clear how soil compaction is connected to the use of chemical fertilizers and bactericides.
Response 2: Thank you very much for your valuable comments. We agree with this comment. Therefore, we have rearranged and better explained this section. We have removed discussions on how soil compaction is connected to the use of chemical fertilizers and bactericides, merely highlighting that soil compaction is a
major challenge for modern agriculture. This change can be found in lines 51-55 of the revised manuscript.
Comments 3: Line 141- what is an appropriate colony density?
Response 3: Thank you for pointing this out. Here, the appropriate colony density refers to having 30-300 colonies on the plate. This range enables clear differentiation of the morphology, size, color, and other characteristics of each colony. As a result, it is conducive to selecting colonies with specific traits for further study. We have revised this section, and this change can be found in line 153 of the revised manuscript.
Comments 4: This can be improved or summarised so that it may read better e.g The antimicrobial activity of mulberry rhizosphere soil bacteria, mulberry endophytic bacteria, and silkworm intestinal bacteria against R. solanacearum, a mulberry bacterial wilt causative pathogen, were determined using the agar well-diffusion
method.
Response 4: Thank you very much for your valuable comments. We agree with this comment. Therefore, we have improved and summarised this section based on your suggestions to enhance readability. This change can be found in lines 175-178 of the revised manuscript.
Comments 5: Line 378- It is not surprising that only 11.1% bacterial strains were capable of nitrogen fixation. Nitrogen fixing bacteria tends to rely mainly on symbiotic relationships in order to promote plant growth by mechanisms such as nitrogen fixation and antimicrobial properties against other competing microbes in
the soil.
Response 5: Thank you for pointing this out. We fully agree with your description of the characteristics of nitrogen-fixing bacteria. Our current study has some limitations regarding the symbiosis of nitrogen-fixing bacteria. We will focus more on this research in the future.
Comments 6: I do appreciate that the authors tried to expand their study as much as possible and produced a significant amount of data, however it would be interesting if a follow up study is done using next generation sequencing technology to identify microbes in the intestines of silkworms that can not be cultured.
Response 6: Thank you very much for your recognition and highly constructive suggestions regarding our research work! You proposed using next-generation sequencing technology to conduct follow-up research to identify microorganisms in the intestines of silkworms that cannot be cultured. This has opened up a new
direction for our research. Indeed, we have recognized the great potential of this technique in addressing current research limitations and have incorporated it into our follow-up research plans. Thank you again for your valuable advice!
Comments 7: I think Line 494-502 is more appropriate in the introduction.
Response 7: Thank you very much for pointing out the content layout in the paper lines 494-502. We fully agree with your view and have moved this section to the introduction. You can find this section in the revised introduction section in lines 70-79. Thank you again for your valuable comments.
Thank you again for your careful guidance.We sincerely hope that the revised manuscript meets your expectations. Should you have any further comments or suggestions on the manuscript, please feel free to let us know. We will take them seriously and make active improvements.

Reviewer 2 Report
Comments and Suggestions for Authors
An interesting and well-described work about the potential of bacterial isolates from mulberry fields toward plant-growth promotion. Especially, the methodology was given in much and proper detail. Please find below my specific comments to further improve the quality of this work.
-Lines 220-230: Please provide more details on the PCR analysis conditions, i.e. the DNA extraction protocol (which kit did you use?), the instrumentation/device models, the database system etc..
-Lines 463-475: Be more critical in the evaluation of the LC-MS findings. You are just giving superficial info on the data. I'd suggest also adding more critical comments to evaluate the importance/meaning of the findings. Can you also include the total ion chromatogram to highlight the abundance of surfactin?
-Lines 571-572: In the genetic analysis, you did not detect the presence of surfactin and iturin as metabolites. However, this was not the case for LC-MS analysis. Please add relevant comments to better clarify this.
-Lines 572-573: You mention that B. licheniformis TF-2 may possess some other
antimicrobial substances in addition to surfactin, iturin, and fengycin. Did you see any other peaks in the total ion chromatogram in LC-MS? You can use these data to provide info on the molecular ion (m/z) values of the unknown compounds and propose their probable molecular composition and/or structure.
Author Response
Dear Reviewer,
Thank you very much for your valuable comments on our manuscript. Your feedback plays a very important role in improving the quality of manuscript and improving the research content. In accordance with your comments, we have made a comprehensive and detailed revision to the manuscript, and now we would like to report the details of the revisions to you point by point.
Comments 1: -Lines 220-230: Please provide more details on the PCR analysis conditions, i.e. the DNA extraction protocol (which kit did you use?), the instrumentation/device models, the database system etc..
Response 1: Thank you very much for your valuable comments. We agree with this comment. Therefore, we have added more detailed information about the PCR analysis conditions in this section. We have included details such as the DNA extraction protocol, the kits used, the instrumentation/device models, and the database system. You can find these additions in Lines 233-250 of the revised manuscript.
Comments 2: -Lines 463-475: Be more critical in the evaluation of the LC-MS findings. You are just giving superficial info on the data. I'd suggest also adding more critical comments to evaluate the importance/meaning of the findings. Can you also include the total ion chromatogram to highlight the abundance of surfactin?
Response 2: Thank you very much for your valuable comments. We agree with this comment. Therefore, we have added more critical comments to evaluate the importance/meaning of the findings, and increased the total ion chromatogram to highlight the abundance of surfactin. This change can be found in lines 490-512 of the revised manuscript.
Comments 3: -Lines 571-572: In the genetic analysis, you did not detect the presence of surfactin and iturin as metabolites. However, this was not the case for LC -MS analysis. Please add relevant comments to better clarify this.
Response 3: Thank you for pointing this out. We agree with this comment. Therefore, we have added relevant comments to better clarify this point. This change can be found in lines 597-600 of the revised manuscript.
Comments 4: -Lines 572-573: You mention that B. licheniformis TF-2 may possess some other antimicrobial substances in addition to surfactin, iturin, and fengycin. Did you see any other peaks in the total ion chromatogram in LC-MS? You can use these data to provide info on the molecular ion (m/z) values of the unknown compounds and propose their probable molecular composition and/or structure.
Response 4: Thank you for pointing this out. We agree with this comment. Therefore, we have added the total ion chromatogram of B. licheniformis TF-2 in this part, and used these data to speculate on the molecular composition and structure of some other antimicrobial substances that B. licheniformis TF-2 may contain. This
change can be found in lines 604-637 of the revised manuscript.
Thank you again for your careful guidance.We sincerely hope that the revised manuscript meets your expectations. Should you have any further comments or suggestions on the manuscript, please feel free to let us know. We will take them seriously and make active improvements.
